# A Recyclable Co-Fe Bimetallic Immobilized Cellulose Hydrogel Bead (CoFeO@CHB) to Boost Singlet Oxygen Evolution for Tetracycline Degradation

Xinying Chen [1,†], He Zhang [1,†], Shizhe Xu [1], Xiaoge Du [1], Kaida Zhang [1], Chun-Po Hu [2,3,4], Sihui Zhan [1], Xueyue Mi [1], Wen Da Oh [5,*], Xiao Hu [2,4,6], Ziyong Pan [7,*] and Yueping Bao [1,*]

1   MOE Key Laboratory of Pollution Processes and Environmental Criteria, College of Environmental Science and Engineering, Nankai University, Tianjin 300350, China
2   Nanyang Environment and Water Research Institute (NEWRI), Nanyang Technological University (NTU), Singapore 637141, Singapore
3   Interdisciplinary Graduate Programme, Graduate College, Nanyang Technological University (NTU), Singapore 637335, Singapore
4   School of Materials Science and Engineering (MSE), Nanyang Technological University (NTU), Singapore 639798, Singapore
5   School of Chemical Sciences, Universiti Sains Malaysia, Gelugor 11800, Penang, Malaysia
6   Temasek Laboratories (TL), Nanyang Technological University (NTU), Singapore 637553, Singapore
7   Tianjin Jinghai District Ecology and Environment Bureau, Tianjin 301600, China
*   Correspondence: ohwenda@usm.my (W.D.O.); 18222891112@139.com (Z.P.); yueping.bao@nankai.edu.cn (Y.B.)
†   These authors contributed equally to this work.

**Abstract:** In the current work, a novel Co-Fe bimetallic immobilized cellulose hydrogel bead (CoFeO@CHB) was prepared via in situ chemical precipitation followed by heat treatment and applied for tetracycline (TC) degradation in the presence of peroxymonosulfate (PMS). The characterization results indicated that the Co-Fe particles were evenly distributed within the porous cellulose hydrogel beads, without affecting their morphologies or crystal structures. During the TC degradation, the CoFeO@CHB/PMS system showed a high resistance and stability to different water bodies, and the common anions and natural organic matters showed a limited effect on TC degradation. The chemical quenching experiments (using chemicals to react with specific reactive species) as well as electron paramagnetic resonance (EPR) results showed that CoFeO@CHB can effectively active PMS to generate multiple reactive oxygen species (ROS, such as $SO_4^{\bullet-}$, $^{\bullet}OH$ and $^1O_2$), in which the $^1O_2$-dominated non-radical pathway played a vital role in TC degradation. Both Co and Fe were proposed as the active sites for PMS activation, and the CoFeO@CHB/PMS system showed a high potential in practical application due to its high selectivity and robustness with much less toxic intermediate products. Furthermore, a long-term continuous home-made dead-end filtration device was constructed to evaluate the stability and application potential of the CoFeO@CHB/PMS system, in which a >70% removal was maintained in a continuous 800 min filtration. These results showed the promising potential for cellulose hydrogel beads utilized as a metal-based nanomaterial substrate for organic degradation via PMS activation.

**Keywords:** hydrogels; peroxymonosulfate; non-radical; tetracycline

## 1. Introduction

As one of the most consumed and produced antibiotics around the world, tetracycline (TC) has been extensively used to prevent and treat infections in both humans and livestock by inhibiting bacterial protein synthesis [1]. It is estimated that 50–80% of applied TC would be excreted into the environment since it cannot be fully absorbed by human and animals, resulting in a large amount of wastewater containing TC being discharged into

various kinds of water bodies [2]. TC can inhibit the growth of biomass and affect the immunity of animals, while the long-term exposure to TC in contaminated water can increase the selection pressure on pathogens and cause the spread the antibiotic resistance genes (ARGs), posing a threat to the ecosystem [3,4]. Furthermore, TC would pose a serious threat to humans via bioaccumulation [5–7]. Therefore, it is of great significance to develop effective methods to remove TC from water.

The advanced oxidation process (AOP) has been widely applied in water/wastewater treatment since it can degrade a variety of non-biodegradable contaminants into benign substances via the generation of reactive oxygen species (ROS) [8]. Sulfate-radical ($SO_4^{\bullet-}$)-based AOP via peroxymonosulfate (PMS) activation has been rapidly developed in recent years due to its high removal efficiencies for various organic pollutants by generating multiple ROS (e.g., $SO_4^{\bullet-}$, $^{\bullet}OH$ and $^1O_2$) [9]. The radical-dominated route ($SO_4^{\bullet-}$ and $^{\bullet}OH$) has been the most widely reported oxidation pathway, in which metal centers ($M^{n+}$), as the active sites, can be oxidized to $M^{n+1}$ while PMS is reduced and decomposed into $SO_4^{\bullet-}$ and $^{\bullet}OH$ [10–12]. The radical-dominated pathway normally has a high oxidative potential (2.5–3.1 V for $SO_4^{\bullet-}$ and 1.8–2.7 V for $^{\bullet}OH$) while the selectivity is relatively low; therefore, it can be easily interfered with by co-existing matter (inorganic anions and organic matter) in water [13]. Meanwhile, the quenching effect caused by reactions between different radicals results in low PMS utilization efficiency and increased operating cost, further limiting the application of radical-dominated catalytic systems [14].

Conversely, the non-radical $^1O_2$-dominated oxidation pathway via PMS oxidation has drawn a lot of attention [15]. As a much milder oxidant, $^1O_2$ is electrophilic and shows a high selectivity to those electron-rich organics [16]. The selective removal of micropollutants reduces the depletion of reactive species by water matrices, therefore, the $^1O_2$-dominated oxidation process exhibits attractive prospects for practical water treatment [14,17,18]. Furthermore, it has been reported that the degradation intermediates via $^1O_2$ showed less toxicity compared to those degraded by the radical-oxidation pathway [12,19]. Therefore, developing a novel $^1O_2$-dominated non-radical oxidation process for TC removal in complex aqueous matrices is meaningful and important.

Currently, the non-radical oxidation pathway via PMS activation has mostly been reported in carbon-based materials, which showed a relatively lower catalytic efficiency and stability [20,21]. Another limitation for the practical application of heterogeneous catalytic systems is the material loss during continuous operation [22,23]. To address these issues, we prepared a novel Co-Fe bimetallic immobilized cellulose hydrogel bead (CoFeO@CHB) via a facile in situ chemical precipitation followed by heat treatment in the current work. The CoFeO@CHB showed a higher catalytic efficiency compared with the pure carbon matrix (CHB), while the easy recyclability of CoFeO@CHB avoided material loss during repeated use. The results showed that the bimetallic oxides were uniformly distributed in the CHB substrate and showed a high catalytic activity for TC degradation via PMS activation. The chemical scavengers experiment and electron paramagnetic resonance (EPR) test evidenced the generation of multiple ROS, in which $^1O_2$ played the dominant role. This $^1O_2$-dominated non-radical oxidation process showed a high level of robustness and stability in multiple water matrices with less toxic intermediates. Furthermore, the long-term performance evaluation in a home-made continuous dead-end filtration device implied that the metal-based nanomaterials can be hybridized with cellulose hydrogel beads to prepare high performance and recyclable catalytic materials for water purification.

## 2. Results and Discussions

### 2.1. Optimization of the Synthesis

The cellulose hydrogel beads with different diameters were first utilized as the substrate to evaluate the effect of substrates on TC degradation. The results in Figure S3 show that the composites with different diameters showed no significant changes in TC removal (>90% removal in 120 min when cobalt was incorporated into the hydrogel beads), while the reaction rate constants were calculated as $0.025 \pm 0.002$ and $0.035 \pm 0.002$ $min^{-1}$ for

2- and 4 mm hydrogel beads, respectively. Moreover, the 4 mm beads were more easily handled during the operation. Therefore, the cellulose hydrogel beads with a diameter of 4 mm showed a higher and less variable rate constant (with a higher $R^2$) and were chosen as the substrates in the current study.

Furthermore, the effect of heat treatment was also investigated. Results in Figure S4 show that the calculated reaction kinetic rate decreased dramatically for different cycles without heat treatment, in which the TC removal efficiency showed the highest reduction when cobalt was incorporated into the cellulose hydrogel beads. The introduction of iron suppressed the cobalt leaching through the interconnection of bimetallic metals, resulting in a higher stability during the treatment. Therefore, the cobalt/iron molar ratio during the synthesis was set as 1:2 in the following sections. Interestingly, the composites were much more stable after heat treatment (150 °C for 3 h) and the calculated reaction rate constant was maintained throughout five cycles (Figure S5). Therefore, it was proposed that the heat treatment strengthened the connection between bimetallic particles and the cellulose hydrogel beads, and was chosen as the optimized synthesis method.

### 2.2. Characterizations of CoFeO@CHB

The photographs of both CHB and CoFeO@CHB are shown in Figure S1, whereby the pristine white cellulose hydrogel beads changed to a little yellow color after the heat treatment (CHB), while the color of CoFeO@CHB was much darker due to the incorporation of CoFeO particles. Figure 1a shows a typical low-magnification TEM image as well as element mapping of the as-synthesized CoFeO@CHB, in which elements of O, Co and Fe are well distributed in the CHB framework. Compared with the pure porous CHB (pore size of several hundred nanometers) in Figure S1, the CoFeO particles with a polydisperse morphology are uniformly dispersed in the CHB framework with a size of around 5 nm without obvious particle agglomeration.

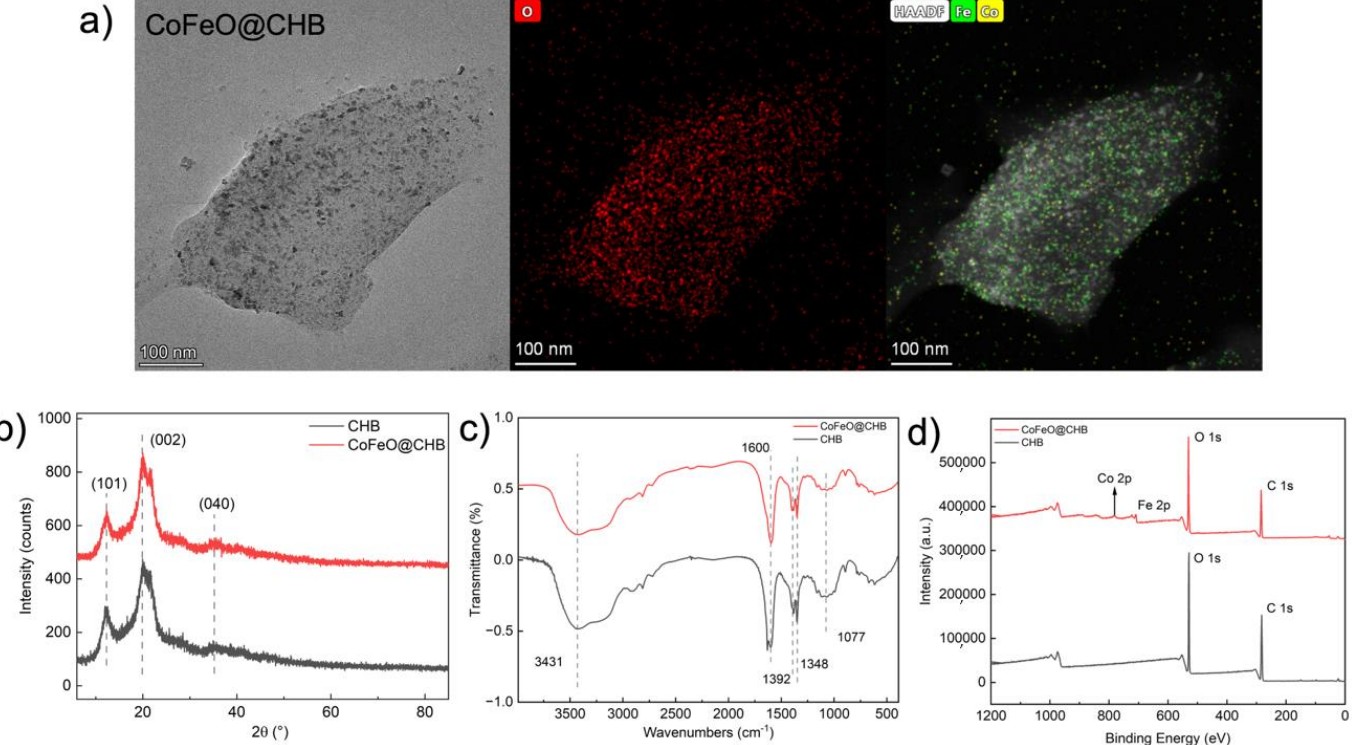

**Figure 1.** Characterizations of CHB and CoFeO@CHB ((**a**)−TEM, (**b**)−XRD, (**c**)−FTIR and (**d**)−XPS).

The crystal phase of both CHB and CoFeO@CHB were confirmed with XRD (Figure 1b), in which the peaks at around 13, 20 and 35° can be identified as (101), (002) and (040)

patterns in cellulose [24]. No characteristic diffraction peaks of cobalt oxide or iron oxides were identified. This could be caused by the relatively low loading amount and weak crystallization. Meanwhile, the good dispersion of the very small particles in the framework was also observed [25]. To calculate the loading amount of CoFeO in CoFeO@CHB, TGA and ICP-MS were applied. As shown in Figure S6, there was no residue for pure CHB after heat treatment at 900 °C because the organic framework of CHB could be burned off completely, while a small residue (0.384%) was found in CoFeO@CHB, which could be used to estimate the CoFeO particle fraction embedded into the cellulose framework. Based on the results from ICP-MS, the loading ratios of Co and Fe in CoFeO@CHB were calculated as 0.07 and 1.72% (atomic ratio), respectively. The significant difference between the Co and Fe ratios indicated that the cellulose hydrogel beads had a higher adsorption affinity to Fe than Co, resulting in the ratio difference from that of the stoichiometry of the initial formulation. From the synthesis cost point of view, the ultra-low loading amount of metals could be cost-effective for the large-scale production.

As shown in Figure 1c, the FTIR spectra of cellulose hydrogel beads showed bands at around 3431, 2900 and 1600 cm$^{-1}$, corresponding to the stretching O-H, C-H and H-O-H vibration [26]. Meanwhile, the band at around 1400 cm$^{-1}$ could be ascribed to the flexing vibration of C-H bond and the band at ~1100 cm$^{-1}$ was identified as a C-O bond in cellulose [27]. No obvious Co-O and Fe-O bands were observed in CoFeO@CHB [28], which could be caused by the ultra-low loading amount.

XPS was further employed to analyze the elemental chemical states and the wide scan in Figure 1d evidenced the coexistence of the elements of C, O, Co and Fe in CoFeO@CHB. Figure 2c shows the high resolution of Co 2p3/2 spectra of CoFeO@CHB and the peaks at around 778.7 and 782.3 eV were assigned to $Co^{3+}$ and $Co^{2+}$, respectively [29,30]. Meanwhile, the peak located at 786.7 eV is the shake-up satellite peak of Co 2p3/2, and $Co^{3+}$ could be formed by the oxidation of $Co^{2+}$ under ambient conditions [31]. Figure S7 displays the Fe 2p spectra of CoFeO@CHB, whereby the peaks located at 708.9 and 722.5 eV were identified as Fe 2p3/2 and Fe 2p1/2, respectively. The peaks at 708.4 and 710.2 eV were associated with $Fe^{3+}$ and $Fe^{2+}$, respectively (Figure 2d) [32,33].

The O 1s XPS spectra of CHB and CoFeO@CHB are shown in Figure 2b. Compared with CHB, a new peak, which was identified as lattice oxygen, was observed in the CoFeO@CHB, evidencing the formation of metal–O bonds [34]. Meanwhile, all the carbon bonds shifted to a lower binding energy because of the incorporation of metals (Figure 2a). All the characterization results confirmed the successful incorporation of bimetallic materials into the cellulose framework.

### 2.3. Catalytic Performance Evaluation

The catalytic performance of as-synthesized CoFeO@CHB was evaluated using TC degradation in the presence of PMS. As shown in Figure S8a, the adsorption of CoFeO@CHB for TC was quite limited during the reaction with a less than 10% removal in 120 min, while TC could be degraded by PMS directly with a nearly 30% removal in 120 min, implying that TC could be oxidized by reactive species generated by PMS self-decomposition [35]. It was observed that TC removal efficiency increased significantly to 76.2% in the presence of both CoFeO@CHB and PMS with a rate constant of 0.0137 ± 0.003 min$^{-1}$, indicating the efficient decomposition of PMS into reactive species in the presence of CoFeO@CHB, resulting in the enhanced degradation of TC. To verify this assumption, PMS decomposition in different systems were investigated and the results are shown in Figure S8b. As indicated, the pure CHB has a quite limited ability to decompose PMS, whereby PMS concentration was maintained at ~90% after 120 min. On the other hand, CoFeO@CHB showed a high reactivity towards PMS decomposition, with an ~30% PMS consumption in 120 min. Interestingly, when TC (10 mg L$^{-1}$) was added into the solution, the decomposition of PMS increased dramatically into >50% in 120 min, which could be caused by the timely consumption of reactive species to TC, further accelerating the decomposition of PMS.

These results showed that the presence of TC may affect the PMS decomposition pathway as well as reaction kinetics.

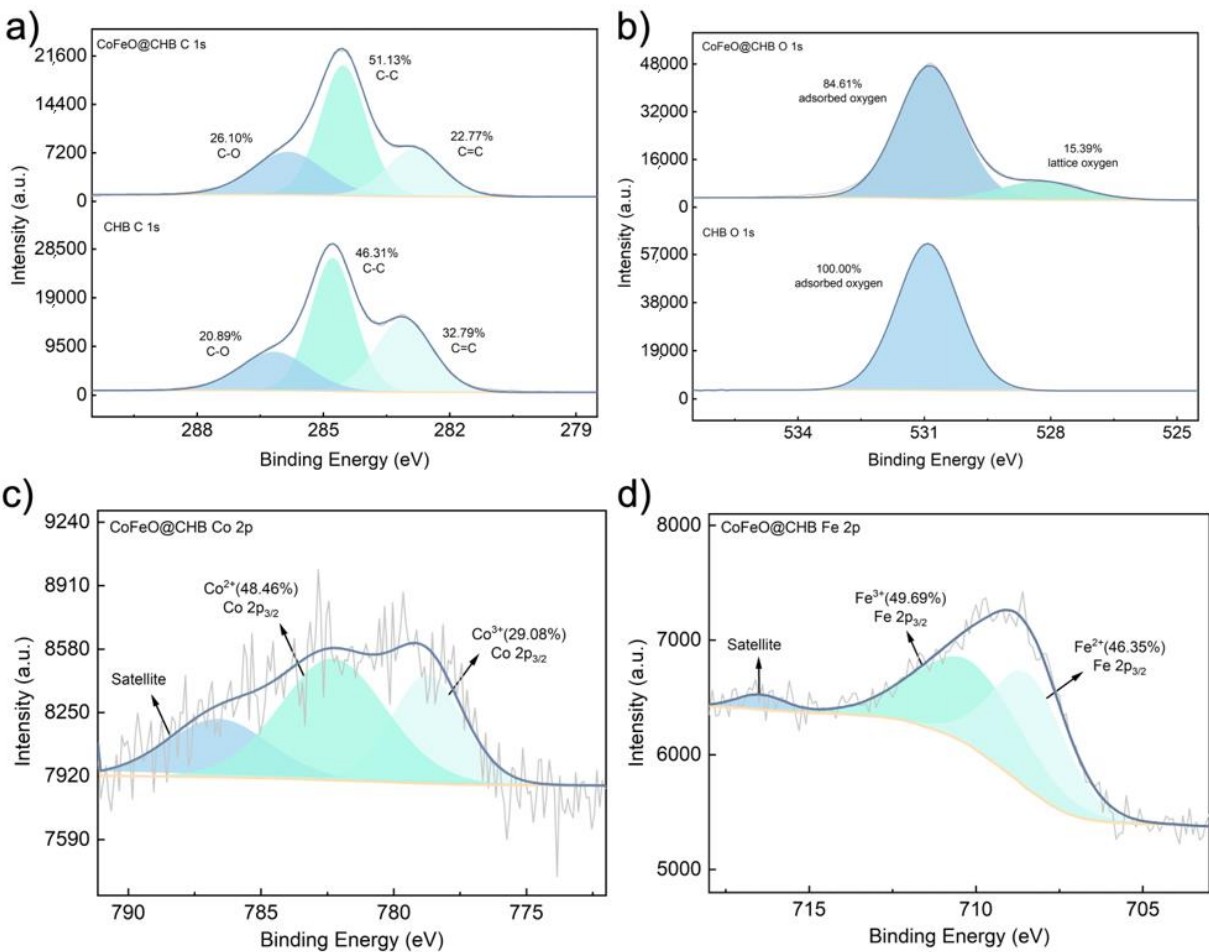

**Figure 2.** High resolution spectra of C 1s (**a**), O 1s (**b**), Co 2p (**c**) and Fe 2p (**d**) in CHB and CoFeO@CHB.

As shown in Figure 3a, when the CoFeO@CHB dosage increased from 0.23 to 1.84 g L$^{-1}$, the TC removal efficiency gradually increased from 63.1 to 79.5% in 120 min with a pseudo-first-order rate constant increase from $0.009 \pm 0.0003$ to $0.0154 \pm 0.002$ min$^{-1}$. There is a positive linear relationship between CoFeO@CHB dosage and reaction rate kinetics (Figure S9a,b). This is understandable, since CoFeO@CHB provided the active sites for PMS decomposition, resulting in considerable reactive species generation and TC degradation [36]. Meanwhile, it was found that the TC removal efficiency increased from 55.2 to 76.2% with the addition of PMS from 0.08 to 0.40 mM (Figure 3b). Consequently, the pseudo-first-order reaction rate constant increased from $0.007 \pm 0.0002$ to $0.014 \pm 0.003$ min$^{-1}$ (Figure S9c,d). The increased PMS dosage would enrich the number of reactive species and result in a promoted TC removal. Interestingly, excessive PMS (0.64 mM) could not further improve the removal efficiency, which could be caused by the scavenging effect between different active species as well as the occupation of active sites in the CoFeO@CHB [37]. From both the economic and environmental aspects, 1.15 g L$^{-1}$ CoFeO@CHB (~200 beads in 1 L) and 0.40 mM PMS were chosen for the following sections.

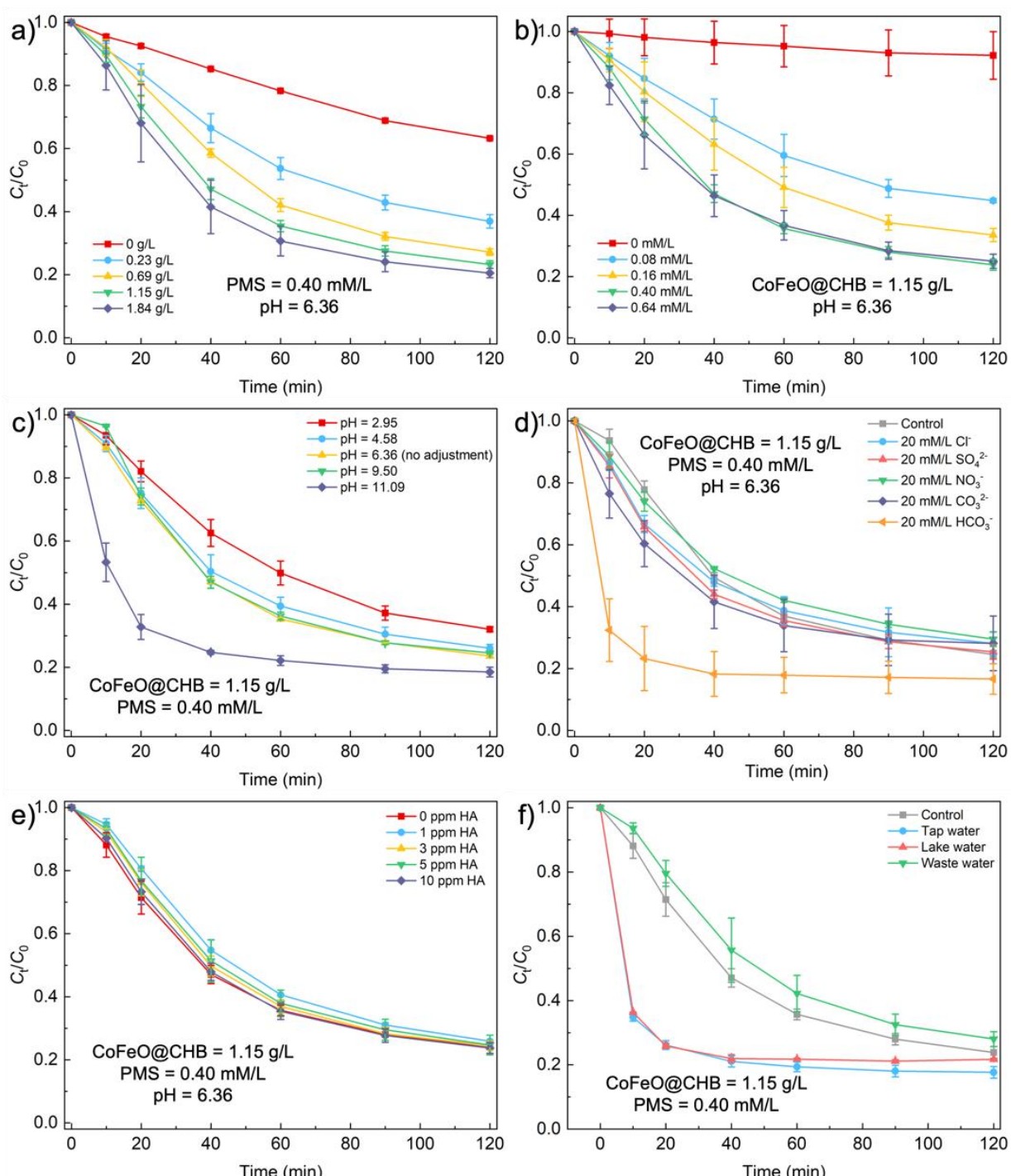

**Figure 3.** Effects of (**a**) CoFeO@CHB loading, (**b**) PMS dosage, (**c**) initial pH values, (**d**) inorganic anions, (**e**) humic acid and (**f**) different water bodies on the degradation of TC.

The pH of the system played a significant role in TC degradation since it can affect the surface property of CoFeO@CHB radical types, as well as existing molecular forms of TC [38]. Therefore, the effect of initial pH values on TC removal in the CoFeO@CHB/PMS system was investigated, and the results are shown in Figure 3c. It can be seen that the removal efficiency increased with pH increase and was maintained at a stable range in pH range of ~5–10. When the initial pH increased from 2.95 to 9.50, the TC removal increased from 68.0 to 75.5% and the rate constant increased from $0.010 \pm 0.0005$ to $0.014 \pm 0.003 \text{ min}^{-1}$. The pKa1 and pKa2 of PMS are <0 and 9.4, respectively, suggesting

that PMS mainly exists in the form of $HSO_5^-$ in acidic or neutral environments and $SO_5^{2-}$ in alkaline environments [39]. At a low pH of 2.95, the O-O bonds in PMS would form strong hydrogen bonds with the $H^+$ in the solution, resulting in low TC degradation efficiency [40]. As the initial pH increased to 4.58–9.50, the reaction rate constant increased (Figure S9e,f). Surprisingly, when the pH increased to a strongly alkaline environment (pH = 11), the degradation capacity increased dramatically, which could be caused by the enhanced self-decomposition of PMS, resulting in the enhanced generation of reactive species, as well as TC removal. Overall, the slight impact of pH variation on TC degradation indicated that the CoFeO@CHB/PMS system has a promising applicability over a wide pH range.

It is well known that there are many inorganic anions as well as organic matter coexisting in natural water environments which may affect the degradation of pollutants. Therefore, a series of experiments about the effects of inorganic anions and humic acid (HA) on TC removal were conducted (Figure 3d,e). It can be clearly seen that the common anions ($Cl^-$, $SO_4^{2-}$, $NO_3^-$, $CO_3^{2-}$) showed no obvious effect on TC removal while $HCO_3^-$ enhanced the degradation process significantly, which could be caused by the increased alkalinity of the solution. On the other hand, humic acid (HA) has been well-reported as a radical scavenger in the natural environment, which may inhibit the organic degradation in the radical-dominated process. However, in the current study, there was no inhibition of TC removal observed in the process when HA concentration increased from 0 to 10 ppm. All these results showed that the CoFeO@CHB/PMS system has a strong resistance to most anions and natural organic matter in natural water, which provides a great possibility for its practical application.

Furthermore, different water matrices, namely tap water, lake water and wastewater, were used to investigate the versatility of the system. As observed in Figure 3f, the degradation of TC increased in both tap water and lake water, and maintained the same performance in wastewater, which could be caused by the slight alkalinity of both the tap water and lake water.

### 2.4. Identification of Reactive Oxygen Species (ROS)

In the common PMS-based advanced oxidation process activated by transition metal oxides, radicals ($SO_4^{\bullet-}$ and $^\bullet OH$) have been regarded as the main reactive species, which would be affected by water matrices [41]. However, in the current work, it can be clearly seen that the TC removal was maintained in different water bodies (Figure 3); therefore, it was assumed that a non-radical pathway induced by $^1O_2$ generation dominated TC degradation in the CoFeO@CHB/PMS system. To verify this assumption, different scavengers (e.g., ethanol for both $SO_4^{\bullet-}$ and $^\bullet OH$, TBA for $^\bullet OH$ and FFA for $^1O_2$) were used to evaluate the contribution of multiple reactive species (Figure 4a). As shown in Figure S10, both ethanol and TBA could not efficiently suppress the TC degradation even at high concentrations (100 mM). This could rule out the main contribution of radicals ($SO_4^{\bullet-}$ and $^\bullet OH$) in TC degradation. However, when 5 mM of FFA was added into the solution, the removal efficiency of TC was reduced to 30%, and 50 mM of FFA fully inhibited the degradation of TC in the system. The relative suppression ratios on TC degradation for 5 mM ethanol, TBA and FFA were calculated as 11.15, 3.50 and 50.33%, respectively (inset of Figure 4a). It has been reported that the FFA may decompose PMS itself and showed the reduced degradation for organics; therefore, EPR was applied to further confirm the formation of $^1O_2$.

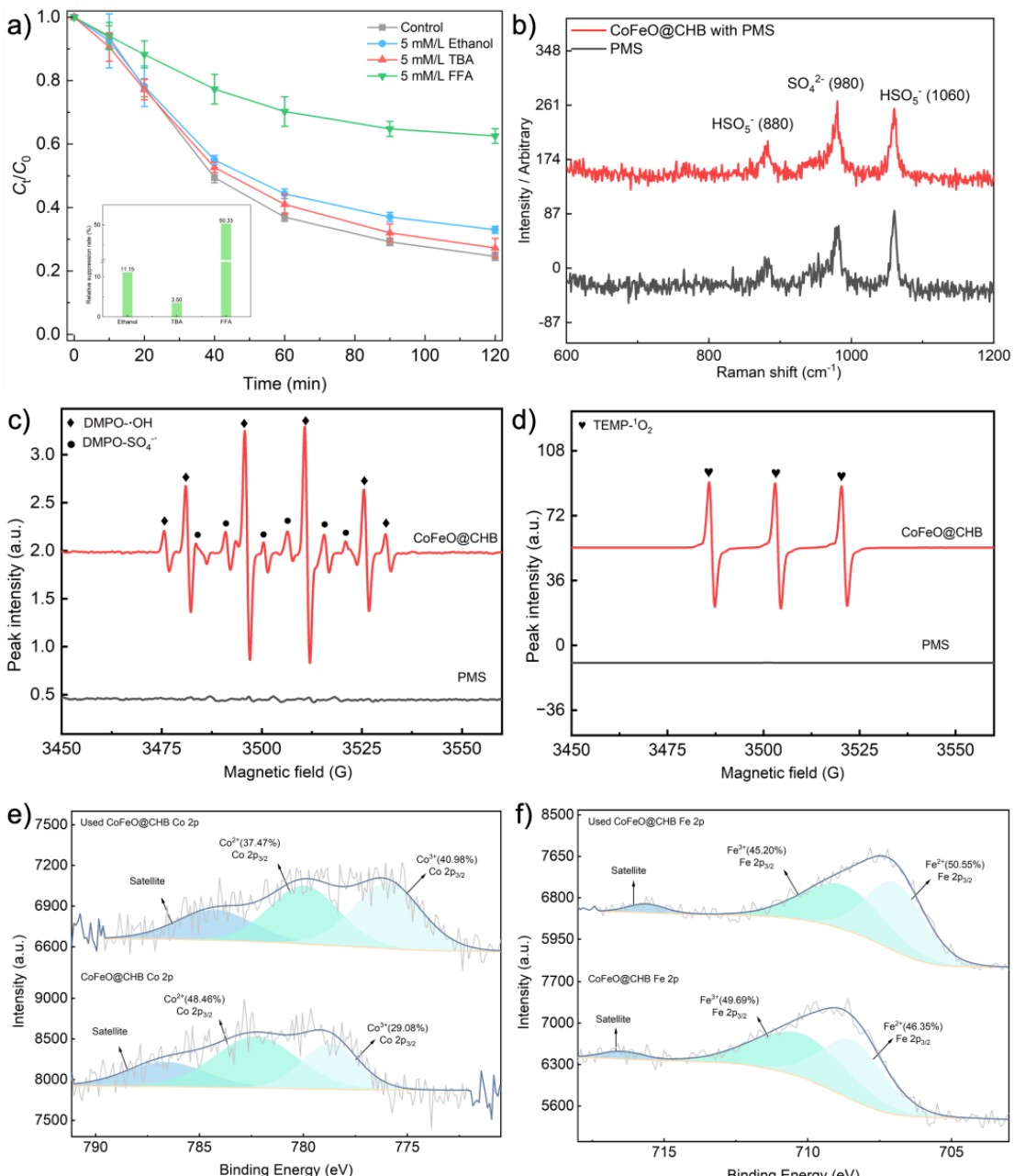

**Figure 4.** Reaction mechanism investigation. (**a**) Reactive species identification via chemical scavengers, (**b**) in situ Raman spectra for both PMS and CoFeO@CHB/PMS, (**c**,**d**) EPR characterizations, (**e**,**f**) high resolution of Co 2p and Fe 2p in both fresh and used CoFeO@CHB.

As shown in Figure 4c and d, no signals of radicals were detected when DMPO/TEMP was added into the solution with only PMS. However, when CoFeO@CHB was added into the reaction solution, the characteristic signals of DMPO-$^\bullet$OH and DMPO-SO$_4^{\bullet-}$ were observed, indicating that CoFeO@CHB could activate PMS to generate both SO$_4^{\bullet-}$ and $^\bullet$OH. Meanwhile, a triplet line with an equal height, which could be denoted as a TEMP-$^1$O$_2$ single, was observed when TEMP was used as the trapping reagent. These characterization results provide solid evidence for the generation of reactive species during PMS activation.

### 2.5. PMS Activation Mechanism and TC Degradation Pathway

To locate the active site in CoFeO@CHB, XPS was applied to both fresh and used CoFeO@CHB and the results are shown in Figure 4e,f. It could be seen that, after the reaction, the ratios of both $Co^{2+}/Co^{3+}$ and $Fe^{2+}/Fe^{3+}$ changed, indicating the electron transfer between the redox reactions (Equations (1)–(4)). As shown in Figure 4e, the ratio of $Co^{2+}$ concentration decreased from 48.46 to 37.47% (10.99%) while the ratio of $Co^{3+}$ concentration increased from 29.08 to 40.98% (11.9%) after the reaction, evidencing the electron donor effect of the Co site. Conversely, the ratio of $Fe^{2+}$ concentration increased from 46.35 to 50.55% (4.15%), accompanied by the decrease in $Fe^{3+}$ concentration from 49.69 to 45.20% (4.49%) (Figure 4f) after the reaction, in which $Fe^{2+}$ could be partially oxidized by $Co^{3+}$ (Equation (5)). Meanwhile, a nucleophilic reaction can occur between $SO_5^{2-}$ and $HSO_5^-$ to generate $^1O_2$ (Equation (6)), and $O_2$ produced from the O-O bonds breakage can be further transferred into $^1O_2$ with the assistance of energy/electron transfer in the system (Equations (7) and (8)).

$$M^{2+} + HSO_5^- \rightarrow M^{3+} + SO_4^{-\bullet} + OH^- \text{ (M = Co and Fe)} \tag{1}$$

$$M^{2+} + HSO_5^- \rightarrow M^{3+} + SO_4^{2-} + {}^\bullet OH \text{ (M = Co and Fe)} \tag{2}$$

$$M^{3+} + HSO_5^- \rightarrow M^{2+} + SO_5^{-\bullet} + H^+ \text{ (M = Co and Fe)} \tag{3}$$

$$SO_4^{-\bullet} + OH^- \rightarrow SO_4^{2-} + {}^\bullet OH \tag{4}$$

$$Co^{3+} + Fe^{2+} \rightarrow Co^{2+} + Fe^{3+} \tag{5}$$

$$HSO_5^- + SO_5^{2-} \rightarrow HSO_4^- + SO_4^{2-} + {}^1O_2 \tag{6}$$

$$2SO_5^{-\bullet} \rightarrow 2SO_4^{-\bullet} + O_2 \tag{7}$$

$$O_2 + \text{energy/electron transfer} \rightarrow {}^1O_2 \tag{8}$$

The lattice oxygen ratio decreased from 15.39 to 5.56% after the reaction (Figure S11d), indicating the loss of metal-O bonds, which could be verified by the metal leaching tests in which the leaching amount of cobalt and iron ions were detected at 0.396 and 0.625 mg L$^{-1}$, respectively. In situ Raman spectra were used to further analyze the affinity between the PMS molecules and the active sites. As shown in Figure 4b, the peaks at 880 and 1060 cm$^{-1}$ were identified as $HSO_5^-$ of PMS, while the peak at 980 cm$^{-1}$ could be associated with the symmetric stretching vibration of the S=O bonds in $SO_4^{2-}$ [42]. The change in the ratio $I_{1060}/I_{980}$ can be used to evaluate the PMS consumption to some extent. As shown in Figure 4b, the $I_{1060}/I_{980}$ of pure PMS was 1.40. However, it decreased to 0.92 in the corresponding CoFeO@CHB/PMS systems, which evidenced the transformation of $HSO_5^-$ into $SO_4^{2-}$.

The TC degradation pathway in the $^1O_2$-dominated system was proposed as Figure 5 and the main intermediate products are identified in Table S1. In pathway I, the C=C bond on the TC ring was broken and -OH was attached to form TC1 ($m/z = 461$) followed by the dehydration reaction via losing hydroxy groups to generate TC3 ($m/z = 427$) and TC5 ($m/z = 404$) [43]. TC4 ($m/z = 406$) and TC6 ($m/z = 366$) were generated from TC3 and TC5 through the deamidation reactions as well as the fourth ring opening. TC4 was further transferred into TC7 ($m/z = 365$) and TC8 ($m/z = 365$) via the oxidation of hydroxy into carbonyl groups. In pathway II, the fourth ring opening occurred at the first step followed by the structure rearrangement to generate TC2 ($m/z = 433$). TC9 ($m/z = 329$), TC10

($m/z = 317$) and TC11 ($m/z = 301$) were generated by oxidation followed by demethylation and dehydration reactions. All the intermediate products with high molecular weight can be further oxidized into smaller ones (TC12-14) and even mineralized into $CO_2$ and $H_2O$. The toxicity of intermediate products was further evaluated via the Toxicity Estimation Software Tool (T.E.S.T., Version 5.1.1), and the results are shown in Figure S12. Most of the intermediates showed a reduced ecotoxicity; some intermediates with higher toxicity could be further oxidized with a prolong reaction time [44], indicating that the CoFeO@CHB/PMS system possessed a high potential in the practical applications.

**Figure 5.** Proposed TC degradation pathway in CoFeO@CHB/PMS system.

### 2.6. Long-Term Performance Evaluation in a Continues Filtration Device

The stability and reusability of catalysts have been identified as a critical parameter for evaluation of materials during the catalytic reaction, and were also investigated in the current work. The recyclability of CoFeO@CHB was investigated using the same materials for TC degradation in five different batches. After each experimental period, the used CoFeO@CHB was washed with DI water and then dried in the oven at 40 °C before next use. As shown in Figure S13a, the degradation performance of CoFeO@CHB could be maintained during five consecutive reactions with a constant $k$ value of around 0.01 min$^{-1}$. Moreover, the physical–chemical properties of CoFeO@CHB after reaction showed no obvious changes, indicating the excellent stability of the materials (Figure S11a–c). We further packed the CoFeO@CHB into a home-made column to evaluate the feasibility and stability for treating synthetic TC wastewater in a continuously flowing reaction (Figure S13b), whereby the removal efficiency can be maintained at >70% with a high-water permeability (~0.12 g min$^{-1}$) in an 800 min filtration.

### 3. Experimental Section

#### 3.1. Chemicals and Materials

The pure cellulose hydrogel beads with a diameter of 2 mm and 4 mm were provided by Rengo Co., Ltd. Japan. PMS ($2KHSO_5 \cdot KHSO_4 \cdot K_2SO_4$) was provided by Sigma-Aldrich (Burlington, MA, USA). TC ($C_{22}H_{24}N_2O_8$, 98%), $Co(NO_3)_2 \cdot 6H_2O$, NaCl (≥99.0%), $Na_2SO_4$ (99.0%), $NaHCO_3$ (≥99.8%), NaOH and HCl were purchased from Shanghai Aladdin Biochemical Technology Co., Ltd. (Shanghai, China) $Fe(NO_3)_3 \cdot 9H_2O$ was provided by Tianjin Damao Chemical Reagent Factory (Tianjin, China). $NaNO_3$ (≥99.0%) was supplied by Tianjin Bohua Chemical Reagent Co., Ltd. (Tianjin, China) and $Na_2CO_3$ (≥99.8%) was purchased from Tianjin Jiangtian Chemical Co., Ltd. (Tianjin, China). Humic acid (HA) was provided by Tianjin Shangya Technology Development Co., Ltd. (Tianjin, China). All

chemicals were used as received without further purification. Deionized (DI) water with a resistance of 18.2 m$\Omega$ cm$^{-1}$ was used in all experiments.

### 3.2. Preparation and Characterization of CoFeO@CHB

The CoFeO@CHB was prepared via a facile in situ chemical precipitation method followed by heat treatment, which was modified from the previous studies [12,45]. As shown in Figure S1, a certain amount of original cellulose hydrogel beads were first immersed into the metal nitrate solution for 6 h with a shaking rate of 200 rpm. Then, the beads were collected and immersed into NaOH solution for 24 h under shaking at 35 °C. After the precipitation, the beads were collected and washed with DI water followed by oven drying at 40 °C. Finally, the beads were heated up to 150 °C in a muffle furnace (SX-G07103) with a heating rate of 5 °C min$^{-1}$ and kept for 3 h. After natural cooling, the beads were collected and named as CoFeO@CHB. CHB was prepared using the similar method in which the metal nitrate precursor solution was replaced by DI water.

The morphology of the materials was observed with a transmission electron microscope (TEM, Talos F200X G2, FEI, Hillsboro, OR, USA). The surface functional groups of the samples were acquired using Fourier Transform Infrared (FTIR, INVENIO, Bruker, Billerica, MA, USA) in a wavenumber range of 400–4000 cm$^{-1}$. Thermogravimetric analysis (TGA) was conducted (GAS Controller GC10, Mettler Toledo, Columbus, OH, USA) to analyze the metal residue of the materials, and the actual metal loading amount was calculated with inductively coupled plasma mass spectrometry (ICP-MS, ELAN DRC-e, PerkinElmer SCIEX, Shelton, CT, USA). The crystal structures of the materials were analyzed using X-ray diffraction (XRD, Rigaku Corporation, Akishima, Japan) in the range of 2θ from 5 to 80° with a Cu Kα radiation (λ = 1.54178 Å), operated at an accelerated voltage of 40 kV and an emission current of 40 mA. X-ray Photoelectron Spectroscopy (XPS, Thermo Scientific K-Alpha, Waltham, MA, USA) was employed to characterize the binding properties of all chemicals. All the XPS spectra were calibrated using C 1s with 284.6 eV at standard peak. In situ Raman spectra were collected on a Raman spectrometer (LabRAM HR Evolution, Horiba, Palaiseau, France) with an excitation wavelength of 532 nm.

### 3.3. Degradation Performance

The degradation of TC was carried out in a batch system with a magnetic stirrer. In a typical reaction, unless otherwise specified, a certain amount of CoFeO@CHB was added into TC solution (10 mg L$^{-1}$). The reaction was then initiated with the addition of PMS (0.40 mM). At the period of time intervals, 3 mL of the solution was collected for detection on a UV–vis spectrophotometer at a wavelength of 357 nm. To investigate the effects of different operating factors, a series of CoFeO@CHB loading (0–1.84 g L$^{-1}$), PMS dosage (0–0.68 mM), pH (3–11), inorganic anions (Cl$^-$, SO$_4^{2-}$, NO$_3^-$, CO$_3^{2-}$, HCO$_3^-$) and HA (0–10 mg L$^{-1}$) was applied to the reaction system. To identify the reactive oxygen species in the system, different chemical scavengers (ethanol, tert-butanol (TBA) and furfuryl alcohol (FFA)) were added into the solution before starting the reaction.

### 3.4. Analytic Methods

The concentrations of TC and PMS during reaction were determined via the UV–vis spectrophotometer (752N, Shanghai INESA Scientific Instruments Co., Ltd., Shanghai, China) at a characteristic wavelength of 357 and 352 nm (mixed with KI solution), respectively [46,47]. The absorbance is positively related to their concentrations in a certain range and the resulting standard curves are shown in Figure S2. The electron paramagnetic resonance (EPR, Bruker A300, Mannheim, Germany) was employed to detect the reactive oxygen species generated in the system, whereby 5,5-dimethyl-1-pyrroline-N-Oxide (DMPO) was used to capture SO$_4^{\bullet-}$ and $^\bullet$OH and 2,2,6,6-tetramethylpiperidine (TEMP) was used as the spin-trapping agent for $^1$O$_2$.

The transformation intermediates during the degradation were analyzed via an UHPLC-Orbitrap-HRMS/MS (Ultimate 3000/Orbitrap Fusion, Thermo Fisher, Waltham,

MA, USA) system. The mass spectrum (*m/z* 70–700) was performed by operating in both positive and negative ion modes. The mobile phase consisted of 0.1% formic acid in either $H_2O$ (A) or acetonitrile (B). The flow rate was 0.4 mL $min^{-1}$ and the column temperature was set at 40 °C with the injection volume of 20 μL.

## 4. Conclusions

A recyclable Co-Fe bimetallic immobilized cellulose hydrogel bead (CoFeO@CHB) was synthesized via in situ chemical precipitation followed by heat treatment. The Co-Fe bimetallic particles were uniformly distributed into the hydrogel matrix, without affecting its crystallinity and morphologies. The CoFeO@CHB/PMS system showed a high robustness and stability during TC degradation, whereby the common anions, natural organic matter and different water bodies had no significant effect on TC removal. More importantly, the reaction rate constant was maintained in five continuous cycles and >70% TC removal could be achieved in the 800 min continuous dead-end filtration process, indicating the potential of the system for practical applications. The novel $^1O_2$-dominated non-radical oxidation pathway transferred TC into the intermediate products with less toxicity. When compared with the literatures (Table S2) [48–54], it was found that the performance of the CoFeO@CHB could be further improved. We proposed that the metal loading could be increased via electrostatic interaction by cellulose carboxylation, which should be considered in future work. Overall, this work provided some insights on the preparation of industrial-scale catalysts by utilizing the natural cellulose hydrogel beads as substrates to achieve high efficiency as well as easy recovery.

**Supplementary Materials:** The following supporting information can be downloaded at: https://www.mdpi.com/article/10.3390/catal13081150/s1, Figure S1: Synthesis and TEM images of as-prepared CHB and CoFeO@CHB; Figure S2: Standard curves for TC (a) and PMS (b); Figure S3: Effects of hydrogel beads diameter on TC degradation (cobalt was loaded into the substrates); Figure S4: Degradation of TC by the (a) first usage, (b) second usage, (c) third usage, (d) fourth usage, (e) fifth usage of catalysts with different Co/Fe ratios and (f) effect of repetition times on reaction rate constants. (Conditions: [TC] = 10 mg $L^{-1}$, [PMS] = 40 mmol $L^{-1}$, [Catalyst] = 1.15 g $L^{-1}$, catalyst prepared without heat treatment); Figure S5: Degradation of TC by the (a) first usage, (b) second usage, (c) third usage, (d) fourth usage, (e) fifth usage of catalysts with different Co/Fe ratios and (f) effect of repetition times on reaction rate constants. (Conditions: [TC] = 10 mg $L^{-1}$, [PMS] = 40 mmol $L^{-1}$, [Catalyst] = 1.15 g $L^{-1}$, catalyst prepared with heat treatment); Figure S6: TGA of both CHB and CoFeO@CHB; Figure S7: High resolution of Fe 2p spectra in CoFeO@CHB; Figure S8: (a) TC degradation in different systems (CoFeO@CHB = 1.15 g $L^{-1}$, PMS = 0.4 mM, pH = 6.36) and (b) PMS decomposition in different systems; Figure S9: The pseudo-first-order kinetic plots for TC degradation and the calculated rate constant under different reaction conditions; Figure S10: Effects of different scavengers on TC degradation in the CoFeO@CHB/PMS system (Conditions: [TC] = 10 mg $L^{-1}$, [PMS] = 40 mmol $L^{-1}$, [CoFeO@CHB] = 1.15 g $L^{-1}$, [pH] = 6.36); Figure S11: The XRD patterns of fresh and used CoFeO@CHB (a), TEM images of used CoFeO@CHB (b), FTIR analysis of fresh and used CoFeO@CHB (c) and high resolution of O1s in fresh and used CoFeO@CHB (d); Figure S12: The toxicity evaluation of TC and its degradation products in the CoFeO@CHB/PMS system; Figure S13: The reusability and stability test of CoFeO@CHB/PMS system for TC degradation in a batch (a) and continuous system (b); Table S1: Determination of TC degradation products in the CoFeO@CHB/PMS system; Table S2. Literature summary of TC degradation by different catalysts/PMS systems.

**Author Contributions:** X.C.: Methodology and formal analysis; H.Z.: Resources and data curation; S.X.: Raman characterization and intermediate products analysis; X.D.: Materials characterizations; K.Z.: Toxicity analysis; C.-P.H. and X.H.: Original cellulose beads supply and materials characterizations; S.Z.: Conceptualization and experimental design; W.D.O. and Z.P.: Conceptualization, Writing—review and editing; X.M.: TEM characterizations and analysis; Y.B.: Conceptualization, Writing—original draft, review and editing. All authors reviewed and edited the original draft. All authors have read and agreed to the published version of the manuscript.

**Funding:** This research was funded by the Fundamental Research Funds for the Central Universities, Nankai University (63231132, 63231195). And the APC was funded by MDPI.

**Conflicts of Interest:** The authors declare no conflict of interest.

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
