# Peer review of "A Recyclable Co-Fe Bimetallic Immobilized Cellulose Hydrogel Bead (CoFeO@CHB) to Boost Singlet Oxygen Evolution for Tetracycline Degradation"

_catalysts, doi:10.3390/catal13081150_

Round 1

Reviewer 1 Report

In MS Chen et al. (catalysts-2502363), results of preparation and catalytic activity of Co-Fe bimetallic immobilized cellulose hydrogel bead is demonstrated for organics degradation (high catalytic activity for teracycline) via peroxymonosulfate activation.

The topic is very important and relevant in many points of view (cellulose polymer-based applications; antibiotic removal; reactive oxygen species; advanced oxidation process), so that it can be interesting for the scientific community and meets the scope of the journal.   

Authors used appropriate routine and advanced experimental techniques (TEM, FTIR, TGA , ICP-MS, XRD, XPS, Raman spectroscopy). The experiments are designed carefully. I do not have anything against the scientific sounds of the data which are provided in the MS. Nevertheless, I have few important, but not serious concerns regarding the presentation.

1)        I suggest defining few “lab jargon like phrases” for non-expert readers (actually, also for experts). Such us: filtration test, filtration experiment, chemical quenching experiments, quenching effect.

 2)        The organization of the MS is not bad. However, I do not think that the data presentations by figures and those of supported materials’ are balanced. There are 7 figures in the MS and 17 figures in supported materials, all are treated almost with equal importance (it appears 7+17=24 figures, which is too much). I suggest focusing on much fewer, however important relevant figures.

 3)        Although, in some figures Transmittance (%) or Intensity (a.u.) is demonstrated, unit scale also should be indicated somehow.

4)        L:278: „… in PMS wound form strong hydrogen bonds with the H+ in the solution…” – „hydrogen bonds” ? Why not protonation/deprotonation? Do we know the pK?

 5)        Interestingly, the error bars on figures are indicated, however, no error values are given for the calculated parameters (kinetic data, % values, etc.) in the text body. Is it possible to supply it?

6)        Please, reconsider using not proper phrases (examples are as follows without completed list):

„… reaction kinetics were calculated as 0.022 and 0.035 min-1…” – „reaction kinetics” ?

„ … more stable reaction rate constant…” ? What does it mean “more stable”?

„ … after the simple heat treatment…” ?

„… to a little yellow color…” ?

„… out obvious aggregation…” ?

„… different water conditions…”

„…Fe2+ increased…”, „…Fe3+ decreased…” 

7)        Regarding style and grammatics.

 Although my English is not native, I think the readability and grammatics of the text can be improved. MS looks grammatically correct, however, there are few grammatic mistakes (see examples below), but literally can be improved.

„… precursor solution was replace by…”

„In-situ Raman  spectra was collected…”

Check the style of the first reference.

 In conclusion, the MS in this form is suitable for publication in a prestigious journal after major corrections and reediting.

Reviewer 2 Report

The manuscript presented work focussing on the bimetallic nanoparticles (Co-Fe) embedded cellulose hydrogel beads for the degradation of tetracycline in the presence of peroxymonosulfate. The work was planned and executed well. However, the authors need to undertake few minor modifications to improve the manuscript.

1.     In Introduction: Apart from “antibiotic resistance genes”, authors are advised to add any other adverse effects possible due to the over presence of tetracycline in the environment.

2.     Please add the reference for the methodology used in the synthesis of CoFeO@CHB (section 2.2. and 2.3.).

3.     Authors should specify the type of reaction kinetics used to determine the rate constant (such as pseudo-first order, pseudo-second order, etc.).

4.     Figure 4: Plot a bar stack graph indicating the overall percentages of different ROS involved in the degradation.

5.     Figure 5d: Make it more presentable.

6.     Figure 7a: Error bars should be included for Cycle 3, 4, and 5 (same as Cycle 1 and 2).

7.     Figure 7b: Include error bars.

8.     Conclusion: Please state as to how to increase the efficiency of the prepared beads, as 120 min is a longer time and recyclability is at ~70% after only 5 cycles.

9.     Provide a comparative table in the degradation of tetracycline through various techniques or by using different nanoparticles embedded beads. This will highlight the need of the present work.

10.  Please include more literature in the discussion of characterization and degradation sections as numerous recent articles are available on this work.

Round 2

Reviewer 1 Report

In MS Chen et al. (catalysts-2502363-v2) is a revised version of MS submitted to Catalysts by the same authors with the same title. After revision there are big improvements in style and conciseness, however I still have concerns.

 Since, the scientific sound is good, I don’t mind publishing it, however, I must stress that I disagree with the authors in few issues.

1)        In my opinion the “supporting information” is “SUPPORTING information”. The MS should be clear for the reader with the information (figures, tables) implemented in the main body of the MS. The contribution of the SI should be more balanced.

2)        In my definite opinion, unit in graphs should be indicated even for percentage scale as well. First of all, because it does exist, it is a scale based on unit definition. In addition, it is important for the reader to know how much it is. E.g., whether it is 0.1, 1.0 or 10, etc. percentage on the scale.

3)        Similarly, unit should be indicated even for arbitrary unit. It is a scale with unit although, arbitryry. Specially, if two or more curves are indicated in the same graph in order to indicate that these curves are in the same magnitude in scale.

4)        I still don’t understand why calculated parameters presented in the text body should not be indicated by errors.

5)        “reaction rate” is not “more stable” it is “with less deviation”, or “less variability”.

6)        I still don’t understand what is the meaning of “simple heat treatment”. Why “simple”?

7)        „.. the ratio of Fe2+ increased…  with the decrease of Fe3+ …”. Correct: the „ratio of Fe2+ concentration” and „decrease of Fe3+ concentration”.

 I advise the authors reconsidering these comments before finalizing the MS.
